# High CCL2 Levels Detected in CSF of Patients with Pediatric Pseudotumor Cerebri Syndrome

**DOI:** 10.3390/children10071122

**Published:** 2023-06-28

**Authors:** Jacob Genizi, Lotan Berger, Muhammad Mahajnah, Yulia Shlonsky, Orit Golan-Shany, Azriel Romem, Ayelet Halevy, Keren Nathan, Rajech Sharkia, Abdelnaser Zalan, Aharon Kessel, Rony Cohen

**Affiliations:** 1Pediatric Department, Bnai Zion Medical Center, Haifa 3104802, Israel; lotan269@gmail.com (L.B.); azrikel@gmail.com (A.R.); keren.nathan@b-zion.org.il (K.N.); 2Bruce Rappaport Faulty of Medicine, Technion, Haifa 3109601, Israel; muhamadmah@hymc.gov.il (M.M.); aharon.kessel@b-zion.org.il (A.K.); 3Pediatric Neurology Unit, Hillel Yaffe Medical Center, Hadera 3810000, Israel; 4Microbiology Lab, Bnai Zion Medical Center, Haifa 3104802, Israel; yulia.shlonsky@b-zion.org.il (Y.S.); orit.golan-shany@b-zion.org.il (O.G.-S.); 5Department of Pediatric Neurology, Schneider Children’s Medical Center of Israel, Petah Tikva 4920235, Israel; drayelethalevy@gmail.com (A.H.); drcohenrony@gmail.com (R.C.); 6Beit-Berl Academic College, Beit-Berl 4490500, Israel; rajachsharkia@hotmail.com; 7Unit of Human Biology and Genetics, Triangle Regional Research and Development Center, Kfar Qari’ 3007500, Israel; dr.zalan@hotmail.com; 8Division of Allergy and Clinical Immunology, Bnai Zion Medical Center, Haifa 3104802, Israel; 9Faculty of Medicine, Tel Aviv University, Tel Aviv 6997801, Israel

**Keywords:** pseudotumor cerebri, cytokines, chemokines, etiology, pediatrics

## Abstract

Pseudotumor cerebri (PTC) is a disorder characterized by increased intracranial pressure in the absence of a structural lesion or other identifiable cause. Cytokines, which are involved in the regulation of immune responses and inflammation, have been implicated in the pathogenesis of PTC. In a prospective, cross-sectional study at three centers in Israel, we analyzed cerebrospinal fluid (CSF) samples from 60 children aged 0.5–18 years, including 43 children with a definitive diagnosis of PTC and a control group of 17 children. Levels of IL-4, IL-10, IL-17, CCL2, CCL7, CCL8, CCL13, BDNF, and IFN-γ were measured using ELISA kits. Levels of CCL2 were significantly higher in the PTC group compared to the control group (*p* < 0.05), with no other significant differences in the measured cytokines between the two groups. The groups did not differ significantly in clinical presentation, imaging, treatment, or ophthalmic findings. Our findings provide preliminary evidence that CCL2 may be involved in the pathogenesis of PTC and may serve a potential target for therapy in PTC.

## 1. Introduction

Pseudotumor cerebri (PTC) is a syndrome consisting of elevated intracranial pressure (250 mm water in adults and 280 mm water in sedated children), a chemically and hematologically normal cerebrospinal fluid (CSF) composition, and papilledema with occasional abducens nerve paresis [1]. PTC is diagnosed following the exclusion of secondary causes of elevated intracranial pressure, such as space-occupying lesions or venous sinus thrombosis, based on appropriate investigations, including brain imaging. The pathogenesis of the raised CSF pressure is still not clear. Some have proposed an increase in CSF production, impairment in CSF absorption, or cerebral edema [2,3,4]. Several studies have proposed that an inflammatory process is implicated in the pathogenesis of PTC [5,6,7,8]. Cytokines are glycoproteins that control inflammation. Elevated levels of inflammatory cytokines, including IL-6, IL-17, IL-1a, and CCL2, were found in a small group of PTC patients [5]. However, those results were inconclusive, requiring further work with a larger patient cohort. Accordingly, our study examined CSF levels of cytokines and chemokines in a larger group of pediatric patients with PTC matched to a control group.

## 2. Materials and Methods

### 2.1. Study Population and Procedure

A prospective, cross-sectional study was conducted in the pediatric departments of three medical centers in Israel: Bnei Zion, Schneider Children’s, and Hillel Yaffe medical centers. Participants were children examined at the centers from 1 January 2020 to 30 December 2022. Children with pseudotumor cerebri, aged 0.5–18 years, were assigned to the study group (CSF was collected before treatment). Children whose clinical symptoms were consistent with increased intracranial pressure and who underwent lumbar puncture (LP), but where PTC was ruled out, served as a control group. An ophthalmologist and a pediatric neurologist conducted a thorough evaluation of all the children. The collected data included a wide range of information, such as clinical symptoms, imaging outcomes, treatment details, ophthalmic examination, and an analysis of the cerebrospinal fluid (CSF). Each hospital obtained its own ethical approval, and the caregivers of all participants provided written informed consent. The local Helsinki committee approved the study.

The CSF was investigated for the following cytokines: IL-4, IL-10, IL-17, CCL2, CCL7, CCL8, CCL13, BDNF, and IFN-γ. The samples were stored at a temperature of −80 degrees Celsius and analyzed at the end of the study. Samples were analyzed using the Luminex xMAP multiplex cytokine assay testing kit—a sandwich enzyme-linked immunosorbent assay (ELISA) kit.

### 2.2. Statistical Analyses

Patients with and without PTC were compared via the chi-square test or Fisher’s exact test, where appropriate, for the categorical data and via independent t-tests for the continuous variables. Odds ratios and their 95% confidence intervals were computed. Significance was considered to be *p* < 0.05. Statistical analyses were performed using SPSS software version 21 (SPSS, Chicago, IL, USA).

## 3. Results

CSF samples from 60 children were collected. Forty three were diagnosed with pseudotumor cerebri, and the other seventeen children in the control group. The distinctive features of the two groups are outlined in Table 1. As noted above, patients in the control group had clinical symptoms consistent with increased intracranial pressure, but their LP opening pressure was normal. The patients in the control group had nonspecific headaches without a chronic headache syndrome or other chronic disease. The patient mean age was 11.8 in the study group and 11.4 in the control group, without statistical differences between them. Patients were more likely to be female, with a male/female ratios of 20/23 and 7/10 for the study group and control group, respectively, again, with no significant differences between them. Clinical symptoms also did not differ significantly between the groups. The opening pressure, as expected, was higher in the study group; please see Table 1.

The Luminex xMAP panel was run for IL-4, IL-10, IL-17, CCL2, CCL7, CCL8, CCL13, BDNF, and IFN-γ. Only CCL2 was detected in the CSF at significant levels, (Figure A1) and it was significantly higher (˂0.03) in the PTC group compared to the control group (Table 2).

## 4. Discussion

We found significantly higher CCL2 levels in the CSF of the pediatric PTC group compared to the control group. This finding was consistent with Dhungana et al.’s [5] finding of increased CCL2 levels in the CSF of adult patients with PTC, based on a small study (eight PTC patients) and using a cytokine antibody array kit. However, they also reported elevated levels of CCL7 and CCL8. In our study, based on 47 pediatric patients with PTC and using a quantitative assay with ELISA, levels of CCL7 and CCL8 were not measurable. On the other hand, Ball et al. [9] found no differences in levels of either CCL2 or other cytokines (IL-1beta, IL-6, IL-8, TNF alpha, hepatocyte growth factor, nerve growth factor, and PAI-1) in patients with idiopathic intracranial hypertension compared to controls.

CCL2 is a chemokine that is involved in the recruitment of monocytes to sites of inflammation. CCL2 activates β1 integrins and regulates the adhesion and chemotaxis of macrophages [10]. Fibroblasts and epithelial cells have the capability to generate CCL2 as well. CCL2 plays a significant role in facilitating the infiltration of monocytes into tissues during inflammatory events. Its expression is induced in response to inflammatory triggers such as infection and tissue injury. The production of CCL2 has been observed in various TH1/M1 inflammatory disorders, including inflammatory diseases. Additionally, CCL2 production is recognized as a characteristic feature of TH2/M2 responses, and it enhances the production of TH2-type cytokines, particularly IL-4, through activated T cells [10]. In the context of PTC, it is thought to play a role in the recruitment of macrophages to the subarachnoid space, leading to inflammation and increased intracranial pressure. It has been suggested that CCL2 may be involved in the regulation of the blood–brain barrier and the production of CSF [11].

Edwards et al. [7], in a small (11 patients) study on adults with PTC, compared levels of IL-17, IL-10, IL-4, and IFN-γ among PTC patients and a control group of patients with other inflammatory diseases (chronic inflammatory demyelinating peripheral neuropathy (CIDP) and multiple sclerosis (MS)). IL-17 was detected among both some PTC patients and patients with MS, but levels of IL-10 were higher among MS patients compared to CIDP and PTC patients. IL-17 is a cytokine produced by T-helper 17 (Th17) cells. T-helper 17 cells play a crucial role in many inflammatory diseases and are involved in recruiting inflammatory cells to the CNS [12]. As such, they may be implicated in the development of PTC as well. IL-10 acts to limit CNS inflammation by modulating the sensitivity of resident glia and infiltrating leukocytes towards activating stimuli, as well as diminishing the production of inflammatory mediators [13]. However, in our study, we found no significant differences in levels of either IL-10 or IL-17 between the PTC and control groups.

We also found no differences in levels of brain-derived neurotrophic factor (BDNF) and IFN-γ between pediatric patients with PTC and the control group. BDNF plays a crucial role in the regulation of neuronal survival, structure, and function, particularly in brain regions responsible for intricate cognitive processes. Studies indicate that disruptions in the signaling pathway of BDNF may contribute to the cognitive deterioration seen in specific neuropsychiatric and inflammatory conditions [14]. Interferon-γ (IFN-γ) is a cytokine produced by T-helper 1 (Th1) cells [15]. Altıokka-Uzun et al. [16], studying adults with idiopathic intracranial hypertension, found elevated levels of TNF-α, IFN-γ, IL-4, IL-10, IL-12, and IL-17 in the CSF of study patients compared to controls. In our study, IL-4, IL-10, IL-17, BDNF, and IFN-γ were all undetected in the CSF of pediatric patients with PTC.

The differences between our findings and those of previous studies may be due to differences in patient populations—specifically, the fact that we looked at pediatric patients with PTC, while all other studies examined adults. There are some known differences between adult and pediatric PTC patients, mainly in the greater prevalence of the male sex and not being obese in young children [17].

The underlying causes of PTC remain undisclosed [18]. While transverse or sigmoid sinus stenosis or atrophy is commonly observed in PTC [1], it is likely a secondary effect rather than the primary cause, resulting from the increased pressure [19]. Nonetheless, stenosis can exacerbate intracranial pressure by impeding the proper removal of cerebrospinal fluid (CSF). The increased prevalence of obesity among PTC patients and the incidence of PTC after specific medicines [1] could potentially be attributed to the impact of estrogen or retinoic acid on epithelial cells, resulting in a reduced CSF outflow. Genetic investigations involving AQP4, which facilitates the movement of water in and out of the brain, have not found any relationship with PTC [20]. Other potential factors to consider include mild inflammation and dysfunction within the glymphatic pathway [21]. In a recent study [18], we explored the role of viruses in the pathogenesis of PTC, but found no supporting evidence. In the present study, we found evidence for a potential role of inflammation (CCL2) in the pathogenesis of PTC. This may support the use of anti-inflammatory therapies, including cytokine-targeted therapies, in the treatment of PTC.

### 4.1. Limitations

Our study, while larger than previous studies (e.g., [5]), was still relatively small, with a small control group; we also did not measure serum levels. We used ELISA, but CCL2 should be confirmed using other methods. Larger studies and ones measuring serum levels are warranted.

### 4.2. Conclusions

Our study provides preliminary evidence that CCL2 may be involved in the pathogenesis of PTC. Additional research is required to validate these findings and explore the therapeutic potential of cytokines as targets for PTC treatment.

## Figures and Tables

**Table 1 children-10-01122-t001:** Characteristics of the study and control groups.

Characteristic	PTC Group (*n* = 43)	Control Group (*n* = 17)	*p*
Mean age	11.8 years	11.4 years	*p* = 0.34
Sex (male/female)	20/23	7/10	*p* = 0.45
Opening pressure (cm H_2_O)	40	24	*p* ˂ 0.001

**Table 2 children-10-01122-t002:** Cytokine levels in CSF of PTC and control groups.

Cytokine	PTC Group (pg/mL)	Control Group (pg/mL)	*p*-Value
IL-10	<42.95	<42.95	0.62
IL-4	<21.59	<21.59	0.53
IL-17	<13.15	<13.15	0.58
CCL2	222	165.2	˂0.03
CCL7	<33.85	<33.85	0.72
CCL8	<13.47	<13.47	0.55
CCL13	<5.51	<5.51	0.46
BDNF	<17.93	<17.93	0.48
IFN-γ	<9.78	<9.78	0.6

## Data Availability

Data is unavailable due to privacy restrictions.

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
