# Peer review of "High CCL2 Levels Detected in CSF of Patients with Pediatric Pseudotumor Cerebri Syndrome"

_children, 2023, doi:10.3390/children10071122_

Round 1

Reviewer 1 Report

In this study the authors investigated cytokine levels in the CSF of pseudotumor cerebri syndrome and found that CCL2 may be involved in the  pathogenesis of pseudotumor cerebri syndrome. Some concerns and suggestions are listed as below:

The sample size of control group (n=17) is low.

In this study, patients in the control group had clinical symptoms consistent with increased intracranial pressure, but their opening pressure was normal. How about the final diagnosis of these patients? Can they serve as a control group?

The exact cytokine levels in CSF of PTC and control groups should be provided.

Cytokin levels in serum of PTC and control groups should be performed.

Were they treated before? I wonder if any treatments have effects on the levels of cytokines.

What about clinical significance regarding increased levels of CCL2?

Increased levels of CCL2 should be confirmed using other methods.

fine

Author Response

Dear Editor

Thank you for your comments and thank you for giving me the opportunity to improve  my manuscript.

Enclosed my response to the comments:

Editor

After your submission, we have checked your manuscript for similarities with former publication and unfortunately the repetition rate is high, according to our guidelines. Please, find the Ithenticate report in the attachment and kindly rephrase highlighted parts to reduce similarities. After you revise, you can use website duplichecker to inspect the manuscript, or you can send it to me and I will gladly check it for you.

Response Thank you for your comment, the paper was extensively revised accordingly. The Ithenticate report was done on the original paper submitted at the 4/5/23 and we also used this version for all the revisions.

Reviewer 1

In this study the authors investigated cytokine levels in the CSF of pseudotumor cerebri syndrome and found that CCL2 may be involved in the  pathogenesis of pseudotumor cerebri syndrome. Some concerns and suggestions are listed as below:

The sample size of control group (n=17) is low.

 Response Thank you for comment. Indeed 17 patients in the control group is not a large group. However, we included only children without inflammatory disease (such as MS), so, recruiting control group who underwent LP and does not have chronic disease is not simple. We mentioned the remark in the limitation section.

In this study, patients in the control group had clinical symptoms consistent with increased intracranial pressure, but their opening pressure was normal. How about the final diagnosis of these patients? Can they serve as a control group?

Response Thank you for your comment. The 17 patients in the control group had non specific headaches without a chronic headache syndrome or other chronic disease. The information was added to the manuscript.

The exact cytokine levels in CSF of PTC and control groups should be provided.

Response Thank you for your comment. The cytokine levels are enclosed in table 2.

Cytokine levels in serum of PTC and control groups should be performed.

Response Thank you for your comment. The cytokine levels are enclosed in table 2.

Were they treated before? I wonder if any treatments have effects on the levels of cytokines.

Response The cytokine levels were done from the CSF  at the initial diagnosis and before treatment.

What about clinical significance regarding increased levels of CCL2?

Response Thank you for your comment, however, the investigation group was not large enough and we could find any correlation between the CCL2 levels and the clinical presentation.

Increased levels of CCL2 should be confirmed using other methods.

Response Thank you for your comment, the limitation was added in the appropriate section.

I hope my clarifications and revisions will make my paper suitable for publication in our journal.

Sincerely

Jacob Genizi

Pediatrics Deportment

Bnai Zion Medical Center, Haifa, Israel.

Tel:  972-4-8359662     Fax: 972-4-8359675                      

e-mail: genizij@gmail.com

Reviewer 2 Report

Dear authors

thank you for submitting this interesting study. Here are some comments:

-Introduction: you have given a lot of information on the epidemiological and clinical diagnosis of PCS. It will also be interesting to add information on the normal and pathological level of cerebrospinal fluid pressure in children as well as data from previous studies on the responsibilities of the autoimmune hypothesis.

-study design: we do not know the precise diagnosis of the clinical symptoms of the control group: what is the cause of this symptomatology which suggests intracranial hypertension

-discussion: too much information is given concerning the function of interleukins when they had a normal value and little discussion on the actual responsibility for the increase in CCL2. The authors state that it may be specific in children but no information if there is indeed a difference in the normal state in children and adults. The conclusion is vague regarding the responsibility for this increase.

-references: ref 2 is duplicated with ref 28

Author Response

Dear Editor

Thank you for your comments and thank you for giving me the opportunity to improve  my manuscript.

Enclosed my response to the comments:

Editor

After your submission, we have checked your manuscript for similarities with former publication and unfortunately the repetition rate is high, according to our guidelines. Please, find the Ithenticate report in the attachment and kindly rephrase highlighted parts to reduce similarities. After you revise, you can use website duplichecker to inspect the manuscript, or you can send it to me and I will gladly check it for you.

Response Thank you for your comment, the paper was extensively revised accordingly. The Ithenticate report was done on the original paper submitted at the 4/5/23 and we also used this version for all the revisions.

Reviewer 2

Thank you for submitting this interesting study. Here are some comments:

-Introduction: you have given a lot of information on the epidemiological and clinical diagnosis of PCS. It will also be interesting to add information on the normal and pathological level of cerebrospinal fluid pressure in children as well as data from previous studies on the responsibilities of the autoimmune hypothesis.

Response Thank you for your comment, the information was added to the introduction.

-study design: we do not know the precise diagnosis of the clinical symptoms of the control group: what is the cause of this symptomatology which suggests intracranial hypertension

Response Thank you for your comment. The main symptoms of the control group was headache and the final diagnosis was non specific headaches. The information was added to the manuscript.

-discussion: too much information is given concerning the function of interleukins when they had a normal value and little discussion on the actual responsibility for the increase in CCL2.

Response Thank you for your comment. We have shortened the discussion on function of interleukins as was in the 4/5/23 submission and added discussion on CCL2.

The authors state that it may be specific in children but no information if there is indeed a difference in the normal state in children and adults. The conclusion is vague regarding the responsibility for this increase.

Response Thank you for your comment. We discussed the differences between PTC in children and adults.  However, this is a preliminary report and it is still too early to make conclusions about the responsibility of CCL2 to PTC in children.

-references: ref 2 is duplicated with ref 28

Response Thank you, Ref 28 was deleted.

I hope my clarifications and revisions will make my paper suitable for publication in our journal.

Sincerely

Jacob Genizi

Pediatrics Deportment

Bnai Zion Medical Center, Haifa, Israel.

Tel:  972-4-8359662     Fax: 972-4-8359675                      

e-mail: genizij@gmail.com

Round 2

Reviewer 1 Report

The authors have answered my questions.

fine